# Viability and Composition Validation of Commercial Probiotic Products by Selective Culturing Combined with Next-Generation Sequencing

**DOI:** 10.3390/microorganisms7070188

**Published:** 2019-06-29

**Authors:** Mati Ullah, Ali Raza, Li Ye, Zhu Yu

**Affiliations:** Hefei National Laboratory for Physical Sciences at Microscale, School of Life Sciences, University of Science and Technology of China, Hefei 230027, China

**Keywords:** probiotics, metagenomics, lactobacillus, bifidobacterium

## Abstract

The consumption of dietary supplements to treat health complications or to improve overall health conditions has become a globally increasing trend that leads to the development of a large number of health-related novel products and expands the associated manufacturing industries around the world. In the current study, we applied selective culturing combined with next-generation sequencing to examine the microbial viability in terms of its culturability on culture medium, composition, and possible contamination in the selected 17 commercial probiotic products sold in the mainland China market. Additionally, the relative abundance of each individual bacterial content was also evaluated by using the generated sequencing reads. The tested probiotic product samples were subjected to Illumina HiSeq-2000 sequencing platform and thoroughly analyzed by the in-house developed bioinformatics pipeline. The comprehensive culturing and sequencing analysis revealed both viability and composition inaccuracy among the several tested probiotic products, however, no contaminant was identified during the analysis. Among the total, five probiotic products (29.41%) were found with an inaccurate or lower colony-forming unit (CFU) counts on culture media while four probiotic products (23.52%) have inaccurately labeled classification. This study provides an ideal qualitative and quantitative assessment approach, which can be used as a diagnostic tool for the accurate assessment of commercial probiotic supplements.

## 1. Introduction

Probiotics, defined by the International Scientific Association of Probiotics and Prebiotic are “the products that deliver live microorganisms with a suitable viable count of well-defined strains with a reasonable expectation of delivering benefits for the well-being of the host” [1,2]. Strains of several bacterial species are used in various health complications that include modulation of immunity, reduction of cholesterol level, Crohn's disease, atopic dermatitis, and diarrhea. Additionally, some studies suggest that some probiotic strains could be effective in rheumatoid arthritis and urinary tract infections [3]. To be considered as probiotic, a product must have to fulfill and meet strict criteria defined for probiotic products that include the quality, safety, and functionality [4]. The two key criteria related to the probiotic products are the viability of the mentioned products contents with a mentioned number of stated cells on the label and the accuracy of the label contents in terms of microbial composition. However, various studies have found significant inaccuracy related to the stated cell numbers and the mentioned compositions on contents labels [4,5,6,7,8].

Strains of several bacterial species are used in the preparation of commercial probiotics products. These include *Bifidobacterium, Lactobacillus, Streptococcus, Pediococcus*, and *Lactococcus* species. The majority of these commercial probiotics products are available in the market in the form of sachet powder, capsule, and gum. The information provided by the supplier on the products label includes the microbial contents and their colony-forming unit (CFU), manufacturing, and expiry date. The popularity of using probiotic products for various health complications has increased recently, and they are used to modulate the individual host immunity against the pathogens, combat their colonization, as well as to maintain the stability the normal human gut microbiota [9,10]. The consumption of probiotic products has been extended from diet supplementation to clinical application in various health complications [10,11]. This alternative application raises serious safety concerns and demands more specific and accurate additional quality control measures.

Over the last few years, several studies have highlighted the inaccuracy in microbial viability and the labeled contents of commercial probiotic products sold in different countries. The microbiological survey of probiotic products contents in the USA has raised a serious concern about the microbial composition, where only four among 13 products were found accurate according to the labels claims [12]. Similarly, another study has pointed the inaccuracy in the viability of the labeled bacteria in 10 out of the total 24 products sold in the European market [13]. Besides this, several other investigations have adopted metagenomic sequencing as a major part of the analysis, combined with selective culturing to investigate the relative abundance of the microbial species found in the probiotic products. These studies have also identified and highlighted the label inaccuracy of microbial contents and the inconsistencies found in viable cell numbers [5,6,8].

The assessment of commercial dietary supplements by high-throughput next-generation sequencing along with selective culturing is an emerging trend aimed to investigate the microbial contents, viability, and any possible contamination found in the probiotic products. In the current study, we applied high-throughput next-generation sequencing combined with a traditional selective culturing technique to validate the label contents, microbial viability, and to examine the presence of any possible contaminants in the several famous commercial probiotic products sold in the mainland China market. We also applied a culture-independent metagenomic sequencing approach to a few products that partially or completely failed to grow on selective solid-growth media. Although culture-independent metagenomic sequence analysis proved to be an ideal analytical tool to quickly analyze the contents and possible contamination found in commercial probiotic products in a single assay, this approach cannot determine the viability of the labeled organisms, which is of key concern. Hence, next-generation sequencing combined with traditional selective culturing can better assess the commercial probiotic products both qualitative and quantitatively.

## 2. Materials and Methods

### 2.1. Probiotic Products

A total of 17 different capsule and sachet-based probiotic supplements were randomly selected and used in this analysis. These products were purchased from several retailers in Hefei, Anhui and stored at 4 °C to maintain the bacterial viability within the end of shelf life. The tested products were from both local and multinational healthcare industries. Prior to the analysis, each of the tested product was renamed in order to keep the anonymity of the products.

### 2.2. Selective Culturing

The majority of probiotic products examined in this study contained a mixture of different bacterial species in a single capsule or powder form enclosed in a sachet. Each capsule or sachet was dissolved in Phosphate-buffered saline (PBS), serially diluted, and plated on general MRS solid plate, as well as on respective selective media according to the claimed organisms. Beside selective media, all the products were also plated on Luria–Bertani (LB) culture medium to rule out the possibility of any gram-negative bacterial contaminants.

### 2.3. Viability and Quantification Assessment

The most important feature of probiotic preparation is the viability and accurate microbial contents mentioned on the label. The viability of the microbial contents was analyzed from the growth results on MRS general-purpose media for *Lactobacillus* and *Bifidobacterium* species and the corresponding selective media for each species. The quantification of bacterial contents was analyzed based on the culture results from the selective media for each bacterial species/strain tested. The quantification of bacterial contents was performed by standard CFU counting method. Briefly, initial dilution of each sample was prepared by mixing one milliliter of the sample with 9 mL sterile PBS. The initial dilution was then serially diluted and mixed with media agar plates to validate the label claim of CFU. One milliliter of the final dilution from the same dilution bank was added to three replicate petri dishes by a pour-plate technique. The plates were gently homogenized and incubated anaerobically at 37 °C for 48 h in an anaerobic incubator (Shanghai Yuejin Medical Instruments Co., Ltd, Shanghai, China). To calculate the final CFU, the resulting colonies were multiplied by the dilution factor and averaged between the replicates.

### 2.4. DNA Isolation

A single isolated colony from pure culture of each selected agar plate was grown from 24 to 48 h on MRS broth, followed by genomic DNA isolation using a Tiangen bacterial DNA isolation kit (Tiangen, Beijing, China), while the QIAamp DNA stool mini kit (Qiagen, Hilden, Germany) was used to isolate DNA directly from probiotics products following manufacturer instructions. The isolated DNA was purified using ethanol precipitation and quantified using Qubit 2.0 fluorometer (Invitrogen, Carlsbad, CA, USA).

### 2.5. PCR for Selected Species

Although the majority of probiotic species were easily distinguished and identified using selective culturing and colony morphology, a few of them (*Lactobacillus casei*, *Lactobacillus paracasei*, *Lactobacillus fermentum*) were not distinguishable in a mixed sample, which was identified through custom-designed specific PCR primers. The primers pair used to differentiate *L. casei*/*paracasei* was LCP1 ACCATCACCAGTGCTGCTAC and LCP2 CAGTGTCCCACTTGGTACCC [6]. For *L. fermentum*, we used species-specific PCR primers couple LF1 AATACCGCATTACAACTTTG and LF2 GGTTAAATACCGTCAACGTA [14]. Similarly, those bacterial species that were not mentioned on label contents but found during growth on solid media were also confirmed by *Lactobacillus* specific and *Bifidobacterium* genus-specific PCR primers. The couples of primers used were LAB 0677F (5′-CTCCATGTGTAGCGGTG-3′), LACT71R (5′-TCAAAACTAAACAAAGTTTC-3′), and BIF-specific (5′-GGTGTGAAAGTCCATCGCT-3′), 23S_bif (5′-GTCTGCCAAGGCATCCACCA-3′), respectively [15,16]. Briefly, a single colony of these products was grown in MRS broth at 37 °C under the anaerobic condition for 24 to 48 h. DNA was isolated using a Tiangen bacterial DNA isolation kit for gram-positive bacteria. The DNA was cleaned via ethanol precipitation and quantified using qubit 2.0. The PCR assay for strain identification was performed with the Bio-Rad Tm100 Touch Thermal Cycler (Bio-Rad, Hercules, CA, USA). The PCR conditions were as follows for all primers denaturation at 95 °C for 2 min; 30 cycles of 94 °C for 30 s, 65 °C for 30 s, and 72 °C for 30 s; and a final extension at 72 °C for 5 min. All PCR products were analyzed by 1% agarose gel electrophoresis.

### 2.6. DNA Library Preparation and Sequencing

A DNA pool was prepared for each probiotic product by mixing all the content DNA in equal quantity prior to library preparation. DNA Libraries were generated using Vazyme TruePrep™ DNA Library Prep Kit V2 for Illumina (Vazyme Biotech, Nanjing, China). The same protocol was used to prepare two libraries from DNA isolated directly from the probiotic products. Sequencing was performed on an Illumina HiSeq-2000 sequencing platform (BGI-Shenzhen, Shenzhen, China) according to the manufacturer’s instruction.

### 2.7. Sequence Analysis

The generated sequencing data were analyzed by in-house-developed bioinformatics pipeline (Figure 1). Briefly, the trimming of the sequencing reads was performed by Trimmomatic [17]. The bioinformatics assembly tools used in the in-house-developed pipeline include COPE [18], SOAPdenovo [19], and Newbler [20]. The assembly of the sequencing reads was processed against the in-house-developed k-mer database that includes a comprehensive collection of sequences related to the species of the normal gut downloaded from NCBI. We downloaded a comprehensive collection of normal gut bacteria from NCBI and identified each tested probiotic bacterial strain against these bacterial species/strain. The data was downloaded to our Linux server on 22 February 2018, to make an in-house database. To avoid the low false-positive rate, the e-value in blast search was set to 1 × 10^−5^. The other parameters of blast search were used as default. The relative abundance of bacterial species was determined by mapping the sequencing reads to the asembled genome. Further, these sequences were converted to small 25 bp segments (25-mer) and were used to calculate the coverage of each individual bacterial species, as described previously [6].

### 2.8. Validation and Confirmation of Sequencing Results

To validate and confirm the sequencing results of all the probiotic products, strain-specific primers were designed based on the sequences of bacterial contents identified during culturing and sequencing results that specifically targeted all of the individual bacteria in each product (Appendix A). The primers were designed by Primer3 with the default setting [21]. Briefly, the individual bacterial colony after selective culturing was grown in MRS Broth at 37 °C for 24 to 48 h, followed by genomic DNA isolation using Tiangen Bacteria DNA kit. The purified DNA was used for the PCR reaction. The PCR conditions for all primers were as follows: initial denaturation at 95 °C for 2 min; 30 cycles of 94 °C for 30 s, 56 °C for 30 s, and 72 °C for 30 s; and a final extension at 72 °C for 5 min. The final PCR products were analyzed by 1% agarose gel electrophoresis.

### 2.9. Validation of Nonculturable Bacterial Contents by PCR

To confirm the bacterial contents mentioned on the products’ labels but not found during culturing (Figure 2), a species-specific PCR reaction was carried out to validate the claimed bacterial contents. The DNA extracted directly from the probiotic products were used as a template for the PCR reaction. The previously described PCR reaction setting, without any modifications for six probiotic bacterial species was used to validate the label composition claim. The adopted specific PCR reaction setting was for *Bifidobacterium longum* [22], *Lactobacillus delbrueckii* sbsp *bulgaricus* [23], *Bifidobacterium lactis* [24], *Streptococcus thermophilus* [25], *Lactobacillus acidophilus* [26], and *Lactobacillus rhamnosus* [27].

### 2.10. Availability of Data

The sequencing raw data used in this study have been deposited in NCBI and allocated the accession number of PRJNA542229.

## 3. Results

### 3.1. Microbial Viability and Quantification

The key features of probiotic products’ preparation are the stable viability and accurate label contents reported during the manufacturing process. To test these claim, each of the individual products was plated on two types of culture media, the general MRS media that usually support the growth of approximately all *Lactobacillus* and *Bifidobacterium* species and the selective media for each microbial contents mentioned on the products label (Table 1). This culturing technique allowed us to determine the microbial viability and count the number of CFU for each bacterial contents in a product. Among the total tested probiotic products, the PB10 was the only product found with no viable bacterial content listed on the label, while the CFU counts of the four others products PB4, PB8, PB9, and PB17 were lower than the label claim (Table 1). The CFU count of probiotic products PB4, PB8, and PB9 was slightly lower than the label claim, while the product PB17 had six bacterial contents found not viable among the total 15, with CFU count determined as 16 × 10^9^ compared with the label claim of 28 × 10^9^ (Table 2). Hence, overall the two products PB10 and PB17 have significant differences among the claimed and determined viable bacterial counts. As the product PB10 had completely failed to grow on culture media while PB17 had nearly half of the bacterial contents found not viable, these two probiotic products were subjected to metagenomics sequence analysis to validate the bacterial composition of the label claim.

### 3.2. Sequence Analysis of Probiotic Products

The summary of the generated sequences from probiotic product samples using Illumina MiSeq next-generation sequencing platforms is summarized in (Table 3). The two probiotic products PB1 and PB3 had low-quality reads and were excluded from the analysis. The number of clean reads was in the range of 5,031,234 to 11,301,642, with mean value 6,951,520, while the GC contents of the generated sequences were in the range between 37.89 to 57.88%, with the mean value of 45.82%. The major part of the analysis (15 samples) was based on the sequences generated from the pool of the DNA isolated from each bacterial contents found during culturing, hence this sequencing approach was solely to confirm our preliminary analysis of culture-based identification, as well as to verify the bacterial contents at species and strain level. This sequencing analysis confirmed the composition discrepancies among the seven probiotic products, including PB2 with major inaccuracy and missing of three labeled bacterial species *Bifidobacterium animalis*, *Bifidobacterium longum*, and *Streptococcus thermophilus* and the presence of three bacterial species *Lactobacillus casei*, *Lactobacillus Plantarum*, and *Lactobacillus rhamnosus* that were not mentioned on the product label (Figure 2). Similarly, this culturing-based sequencing approach also confirmed some additional species found in three probiotics products PB7, PB14, and PB15, each of which contained one or two bacterial species not mentioned on the products label (Figure 2). The probiotic product PB7 contained *Lactobacillus casei* as an additional bacterial content while missing the *Bifidobacterium longum* mentioned on the label; similarly, the PB14 missed the two labeled species *Bifidobacterium animalis* and *Lactobacillus rhamnosus* and contained *Bifidobacterium longum* and *Lactobacillus helveticus* instead. The product PB15 was also found with label inaccuracy, where both the labeled bacterial species *Bifidobacterium longum* and *Lactobacillus acidophilus* were missing while two other Lactobacillus species that are *Lactobacillus rhamnosus* and *Lactobacillus plantarum* were present instead (Figure 2).

The majority of health benefits associated with probiotic products are specific to a bacterial species or strain level and can be significantly different from the members of the same species or different strains. Our culture-based next-generation sequencing provided strain-level information for all the bacterial contents reported in each individual probiotic product (Table 4). Similarly, the sequencing analysis also provided the relative abundance of each bacterial content in a mixed sample product (Figure 3). This highlights the advantage of our approach for the strain-level identification of bacterial contents in a mixed population.

The metagenomic sequence analysis of two probiotic products PB10 and PB17 validated the bacterial label contents of both the products (Figure 2). The probiotic product PB10 had four bacterial species mentioned on the product label, but none was identified viable during culturing techniques. Similarly, PB17 had 15 bacterial species mentioned on the product label, and only nine were identified viable during culturing analysis (Figure 2). Subjecting these two products to metagenomics sequence analysis accurately validated the claimed label composition.

Numerous bacterial species were identified to comprise the preparation of probiotic products. These bacterial species have generic health benefits, as well as being used for the targeted therapy in various health complications. The survey of these products was based on a random collection of different commercial probiotic products sold in the mainland China market from both local and multinational manufacturing health industries. Based on our analysis, a total of 18 bacterial species were identified that have comprised one or more probiotic products tested in this study. Among the total bacterial species identified, *L. acidophilus* was found with the highest prevalence and was present in ten probiotic products, followed by *L. rhamnosus* and *B. animalis* that were identified in nine and seven probiotic products, respectively (Figure 4).

### 3.3. Confirmation of Sequence Analysis

The additional confirmation of sequencing results was carried out by a polymerase chain reaction. Strain-specific primers were designed based on the assembled sequences for each bacterial contents (Appendix A). The final PCR products were examined on gel electrophoresis to verify the targeted amplified sequence. All of the tested primers have shown the desired amplified PCR product for each of the tested bacterial contents.

### 3.4. Validation of Nonculturable Probiotic Bacterial Contents

Other than the two probiotic products PB10 and PB17 that were subjected to metagenomics sequence analysis, the confirmation of all others bacterial contents found nonculturable during culturing and not included in the sequencing analysis were assessed by species-specific PCR reaction settings established previously. A total of six bacterial species in seven probiotic products were analyzed, among which four bacterial species in two probiotic products were identified with PCR and confirmed the label accuracy, while none of the bacterial species in the remaining four probiotic products were detected in PCR confirmation reaction and were concluded as missing or having inaccurate labeling (Figure 2).

## 4. Discussion

The consumption of dietary supplements for health benefits has a long history, but over the recent several years this trend has emerged significantly around the globe. The use of probiotic bacteria for improving overall health condition as well as to cure different illnesses has been widely reported among people of all age groups. This has resulted in the development of large numbers of new probiotic products with different claimed benefits from hundreds of different health industries. However, according to the most recent definition of probiotics by World Health Organization (WHO) and the International Scientific Association of Probiotics and Prebiotics, the probiotic is the product that contains live microorganisms with well-defined strains and with adequate amounts to deliver health benefits to the host [2]. By this definition, the two important features of probiotic products are their viability and the desired number required for delivering specific effects on host health. In the current study, we have comprehensively analyzed 17 different probiotics products (both from local and multinational manufacturers) sold in the mainland China market to validate the label claim. Keeping in mind the FAO/WHO recommendations regarding probiotics, we adopted an analytical approach composed of selective culturing combined with next-generation sequencing to deeply analyze the microbial contents of these commercial probiotics products. Herein, we called this approach culturing-based sequencing that can validate both the bacterial viability as well as the composition of probiotic products at species and strain level.

Of note, culture-based sequencing can only be applied to microbial contents that show growth on a culture medium, hence for cells without growth, we applied two different strategies to validate the composition of the label claim. This included metagenomics sequencing, in which case the DNA isolated directly from the probiotic products were subjected to high-throughput sequencing and species-specific PCR reaction that can specifically identify the claimed species. Although a positive PCR result may rule out the absence of the cells, it is not a sufficient proof to conclude on their nonviability, as factors such as death, absence, or a switch to a nonculturable phenotype may influence cells nonculturability. The metagenomic sequencing approach was applied to two products, PB10 and PB17 that accurately validated the composition of label claim. The product PB10 was the only product that completely failed to show viability on a culture medium, however, all of the four mentioned species sequences were recovered using metagenomic sequence analysis. Similarly, the product PB17 had the highest number of bacterial species mentioned on the label (15 species), and nearly half of the bacterial species were found not viable on culture medium. The metagenomics sequence analysis of this product validated all of the 15 bacterial species mentioned on the product label (Figure 2). This highlights the strength of metagenomics analysis for rapid identification of bacterial community where the viability of the microbiota is not important. However, in the case of probiotics products investigation, using this technique alone cannot provide the desired assessment.

The use of next-generation sequencing for the assessment of commercial probiotic products is an emerging trend, and several studies have highlighted its significance recently and over the last few years [5,6,8]. Using next-generation sequencing as a basic tool, different strategies were employed to investigate and validate the label contents of probiotic products. All of the mentioned reports have highlighted significant discrepancies in the bacterial contents and their viability mentioned on the product labels [5,6,8]. However, these studies have used next-generation sequencing as a primary tool that primarily aims to investigate the product contents via a metagenomic sequencing approach, and the culturing of the microbial contents was carried out as a secondary verification tool. In contrast, we combined the next-generation sequencing with selective culturing for the major part of our analysis to validate the commercial probiotic products both qualitative and quantitatively.

The culturing strategy adopted here for the validation of bacterial viability and estimation of desired CFU counts has enabled us to identify bacterial species not listed on the product label. As all of the individual probiotic product contents were plated on general MRS and species-specific selective culture media (Table 1) to examine the bacterial viability and desired growth counts, the unlisted bacterial species were easily identified from the culture growth on MRS general-purposes media. Although, these bacterial species were confirmed by species-specific PCR reaction prior to sequencing. The use of MRS media for probiotic bacterial culturing and enumeration has been widely reported and is considered one of the most ideal medias for probiotic *Lactobacillus* and *Bifidobacterium* species identification [28,29].

The knowledge of strain-specific health benefits of probiotic products is important for several consumer groups that use probiotic supplements as targeted therapy. Although the majority of the probiotic products analyzed in this study have labeled information at both the species and strain level, a few of the tested products have not provided the strain-level information on the product label. It is therefore important to screen the bacterial contents associated with the probiotic properties at strain level to make it a clear and precise choice for consumer consumption. Our culture-based sequencing has successfully identified each of the probiotic contents at the strain level for all of the tested products (Table 4).

In summary, we have introduced a culture-based sequencing approach that can comprehensively analyze and validate the microbial contents of commercial probiotic products both qualitative and quantitatively. Using this approach we have validated and identified both qualitative and quantitative discrepancies in several famous commercial probiotic products sold in the mainland China market. Additionally, this approach has provided the strain-level information for all of the tested probiotic products’ contents, confirming the ability of validation of “well-defined strain” criteria of commercial probiotic products. Finally, we would like to suggest that the manufacturer should provide some additional information to the customers on the product label or website about the methodology used in the quality-control measurements, especially concerning bacterial viability and microbial purity. This information can be helpful to assess the quality of probiotic products in a much broader range.

## Figures and Tables

**Figure 1 microorganisms-07-00188-f001:**
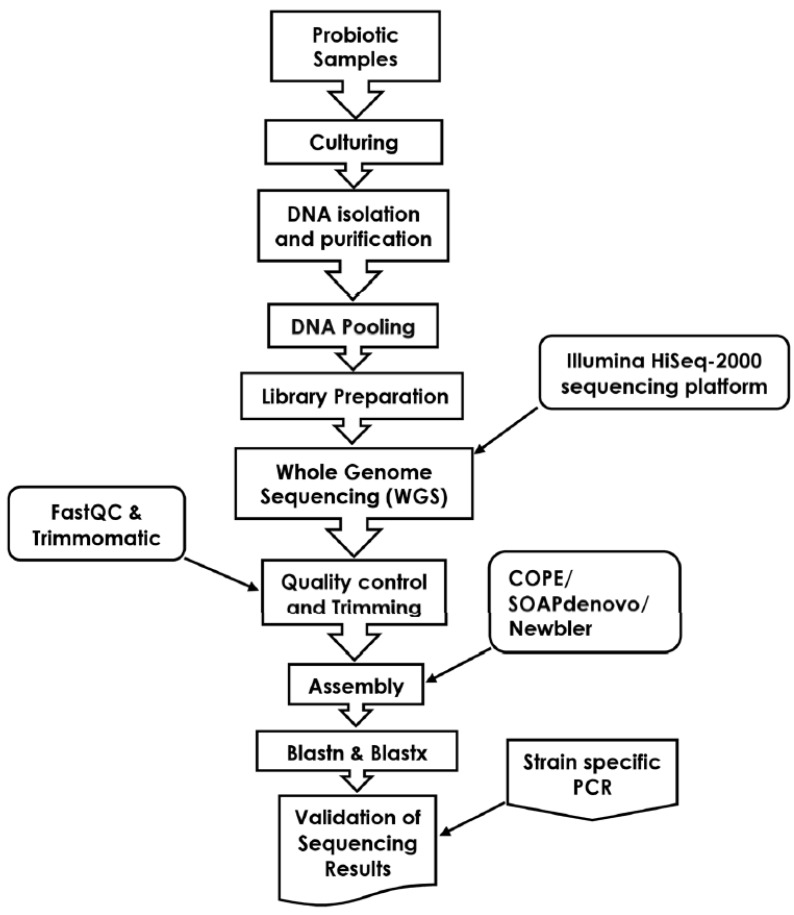
Schematic representation of culture-based sequencing analysis of commercial probiotic products.

**Figure 2 microorganisms-07-00188-f002:**
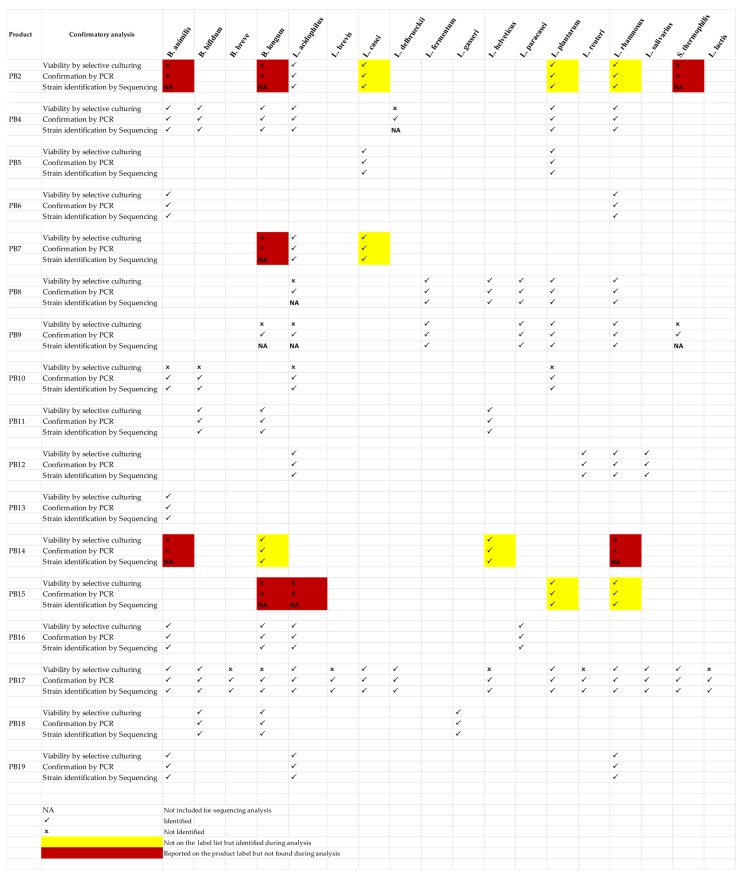
Summary of the qualitative analysis of probiotic products and identified discrepancies.

**Figure 3 microorganisms-07-00188-f003:**
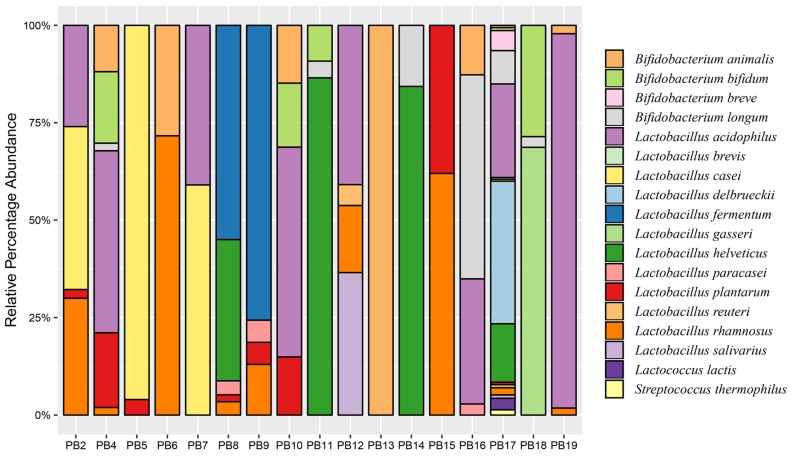
Relative percentage abundance of probiotic bacterial contents in each product.

**Figure 4 microorganisms-07-00188-f004:**
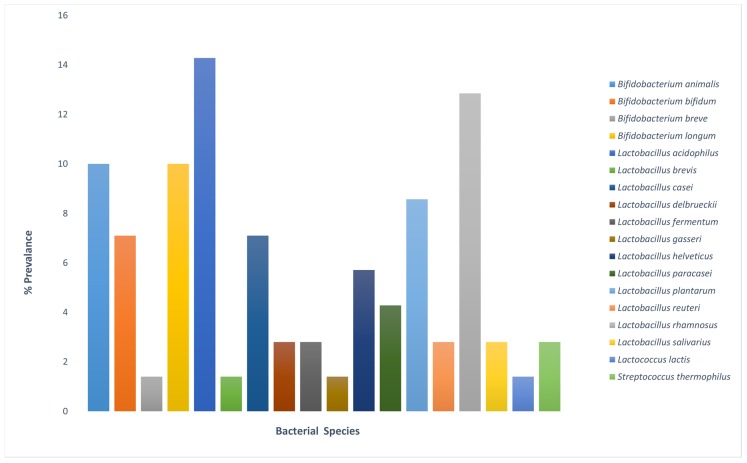
Percentage prevalence of each bacterial species identified in the tested probiotic products.

**Table 1 microorganisms-07-00188-t001:** Growth media and culturing conditions used for probiotics products’ contents.

Bacteria	Medium	Growth Condition
*Bifidobacterium* spp.	MRS/ MRS-NPNL	anaerobic incubation at 37 °C for 72 h
*Lactobacillus acidophilus*	MRS-clindamycin	anaerobic incubation at 37 °C for 72 h
*Lactobacillus casei*	MRS-Bile	aerobic incubation at 37 °C for 48 h
*Lactobacillus delbrueckii*	MRS 5.2	pH 5.2, anaerobic incubation at 45 °C for 72 h
*Lactobacillus fermentum*	MRS	anaerobic incubation at 37 °C for 48 h
*Lactobacillus gasseri*	MRS	anaerobic incubation at 37 °C for 48 h
*Lactobacillus helveticus*	MRS	aerobic incubation at 37 °C for 48 h
*Lactobacillus paracasei*	MRS-Bile	aerobic incubation at 37 °C for 48 h
*Lactobacillus plantarum*	MRS	anaerobic incubation at 37 °C for 48 h
*Lactobacillus reuteri*	MRS	aerobic incubation at 37 °C for 48 h
*Lactobacillus rhamnosus*	MRS Vancomycin	anaerobic incubation at 37 °C for 48 h
*Lactobacillus salivarius*	MRS	anaerobic incubation at 37 °C for 48 h
*Streptococcus thermophilus*	M17 Agar	aerobic incubation at 37 °C for 48 h
*Lactococcus lactis*	M17 Agar (0.5% glucose)	aerobic incubation at 30 °C for 48 h

**Table 2 microorganisms-07-00188-t002:** Comparison of product labeling values for viability versus bacteriological culturing conducted in this study.

Probiotic	Label Composition ^a^	CFU Count ±SD ^b^
PB2	8×10^9^ CFU of *L. acidophilus, B. animalis*, *B. longum, S. thermophilus*	8×10^9^ ± 0.075
PB4	9×10^9^ CFU of *L. rhamnosus, B. bifidum, B. animalis, L. acidophilus S. thermophilus, L. bulgaricus*, *B. longum*	7.5×10^9^ ± 0.012
PB5	2.0 × 10^9^ CFU *L. casei*, *L. plantarum*	2.0 × 10^9^ ± 0.078
PB6	30×10^9^ *B. animalis*, 30×10^9^ *L. rhamnosus*	60 × 10^9^ ± 0.080
PB7	*L. acidophilus* 5.3 × 10^9^, *B. longum* 1 × 10^9^	6.3 × 10^9^ ± 0.0318
PB8	6×10^9^ CFU of *L. acidophilus, L. fermentum, L. helveticus, L. paracasei, L. plantarum, L. rhamnosus*	5.0 × 10^9^ ± 0.059
PB9	17.5 × 10^9^ CFU of *B. longum, L. acidophilus, L. fermentum, L paracasei, L. plantarum, L. rhamnosus, S. thermophilis*	10.5 × 10^9^ ± 0.013
PB10	1×10^7^ CFU of *L. acidophilus, B. lactis, B. bifidum, L. planatarum*	NG
PB11	1.425 × 10^9^ CFU *B. bifidum*, 1.425 × 10^9^ CFU *B. longum (infantis)*, 9.6 × 10^9^ CFU *L. helviticus*	12.4 ×10^9^ ± 0.005
PB12	10 × 10^9^ CFU of *L. reuteri, L. salivarius, L. acidophilus, L. rhamnosus*	10 × 10^9^ ± 0.033
PB13	1x10^9^ CFU of *B. animalis*	1 × 10^9^ ± 0.055
PB14	6× 10^9^ CFU of *B. animalis*, *L. rhamnosus*	6× 10^9^ ± 0.015
PB15	5× 10^9^ CFU of *L. acidophilus, B. longum*	5× 10^9^ ± 0.013
PB16	4 × 10^9^ CFU of *L. acidophilus, B. longum, B. animalis, L. paracasei*	4 × 10^9^ ± 0.071
PB17	28 × 10^9^ CFU of *L. acidophilus, B. longum, B. bifidum, B. breve, B. lactis, L. bulgaricus, L. casei, L. helviticus, L. plantarum, L. reuteri, L. rhamnosus, L. salivarius, S. thermophilus*, *L. brevis, L. lactis*	16 × 10^9^ ± 0.028
PB18	4 × 10^9^ CFU of *L. gasseri, B. bifidum, B. longum*	4 × 10^9^ ± 0.072
PB19	3 × 10^9^ of *L. acidophilus, B. animalis, L. rhamnosus*	3 × 10^9^ ± 0.063

^a^ Information labeled on probiotics products, ^b^ Determined in this study, The values represent the average of three replicates along with the standard deviation, NG: No growth. CFU: colony-forming unit.

**Table 3 microorganisms-07-00188-t003:** Summary of the sequencing data generated from probiotic products samples.

Sample	Raw Reads	Clean Reads	Raw Base(G)	Clean Base (G)	GC%
PB2	19,956,035	5,410,890	5.99	1.62	44.66
PB4	8,399,739	6,419,867	2.52	1.93	46.55
PB5	9,661,983	7,548,271	2.9	2.26	46.39
PB6	9,194,286	7,793,312	2.76	2.34	48.39
PB7	8,835,931	6,599,652	2.65	1.98	43.4
PB8	8,193,811	5,031,234	2.46	1.51	46.44
PB9	7,913,048	6,447,026	2.37	1.93	50
PB10	9,669,794	6,868,151	2.9	2.06	45.11
PB11	9,486,662	7,029,632	2.85	2.11	42.34
PB12	11,201,502	8,544,999	3.36	2.56	39.22
PB13	13,010,708	11,301,642	3.9	3.39	57.88
PB14	13,461,984	9,517,061	4.04	2.85	43.04
PB15	9,865,374	7,615,037	2.96	2.29	46.2
PB16	8,262,088	2,411,023	2.48	0.72	52.7
PB17	10,252,995	8,199,392	3.08	2.46	46.2
PB18	7,823,958	5,023,448	2.35	1.51	42.55
PB19	8,257,723	6,415,206	2.48	1.93	37.89

**Table 4 microorganisms-07-00188-t004:** Strain-level identification of probiotic products’ bacterial contents.

	Bacterial Contents Mentioned on the Product Label	Bacterial Contents Identified during Sequence Analysis
PB2	Not Mentioned	*Lactobacillus rhamnosus* LOCK908
	Not Mentioned	*Lactobacillus casei* W56
	Not Mentioned	*Lactobacillus plantarum* strain B21
	*Lactobacillus acidophilus* strain FS14	*Lactobacillus acidophilus* strain FSI4
PB4	*Lactobacillus rhamnosus* ATCC 53103	*Lactobacillus rhamnosus* ATCC 53103
	*Bifidobacterium bifidum* ATCC 29521	*Bifidobacterium bifidum* ATCC 29521
	*Lactobacillus acidophilus* strain FSI4	*Lactobacillus acidophilus* strain FSI4
	*Bifidobacterium animalis*	*Bifidobacterium animalis* strain A6
	*Bifidobacterium longum*	*Bifidobacterium longum* subsp. *infantis* ATCC 15697
PB5	*Lactobacillus casei*	*Lactobacillus casei* W56
	*Lactobacillus plantarum* ZS2058	*Lactobacillus plantarum* ZS2058
PB6	*Lactobacillus rhamnosus* LOCK908	*Lactobacillus rhamnosus* LOCK908
	*Bifidobacterium animalis* sbsp. *lactis*	*Bifidobacterium animalis* sbsp. *lactis* strain A6
PB7	Not Mentioned	*Lactobacillus casei* BL23
	*Lactobacillus acidophilus* La-14	*Lactobacillus acidophilus* La-14
PB8	*Lactobacillus paracasei*	*Lactobacillus paracasei* strain L9
	*Lactobacillus rhamnosus* LOCK908	*Lactobacillus rhamnosus* LOCK908
	*Lactobacillus plantarum* subsp. *plantarum* ST-III	*Lactobacillus plantarum* subsp. *plantarum* ST-III
	*Lactobacillus helveticus* R0052	*Lactobacillus helveticus* R0052
	*Lactobacillus fermentum* 3872	*Lactobacillus fermentum* 3872
PB9	*Lactobacillus rhamnosus* ATCC 53103	*Lactobacillus rhamnosus* ATCC 53103
	*Lactobacillus plantarum* LZ95	*Lactobacillus plantarum* strain LZ95
	*Lactobacillus paracasei* L9	*Lactobacillus paracasei* strain L9
	*Lactobacillus fermentum* 3872	*Lactobacillus fermentum* 3872
PB10	*Lactobacillus plantarum* B21	*Lactobacillus plantarum* strain B21
	*Bifidobacterium bifidum*	*Bifidobacterium bifidum* ATCC 29521
	*Lactobacillus acidophilus* La-14	*Lactobacillus acidophilus* La-14
	*Bifidobacterium animalis subsp lactis*	*Bifidobacterium animalis* strain A6
PB11	*Lactobacillus helveticus* R0052	*Lactobacillus helveticus* R0052
	*Bifidobacterium bifidum* BGN4	*Bifidobacterium bifidum* BGN4
	*Bifidobacterium infantis* subsp. *infantis*	*Bifidobacterium longum* subsp. *infantis* ATCC 15697
PB12	*Lactobacillus rhamnosus* 53103	*Lactobacillus rhamnosus* ATCC 53103
	*Lactobacillus reuteri*	*Lactobacillus reuteri* SD2112
	*Lactobacillus acidophilus*	*Lactobacillus acidophilus* strain FSI4
	*Lactobacillus salivarius*	*Lactobacillus salivarius* strain *Ren*
PB13	*Bifidobacterium animalis*	*Bifidobacterium animalis* strain A6
PB14	Not Mentioned	*Lactobacillus helveticus* R0052
	Not Mentioned	*Bifidobacterium longum* subsp. *infantis* 157F
PB15	Not Mentioned	*Lactobacillus rhamnosus* ATCC 53103
	Not Mentioned	*Lactobacillus plantarum* strain B21
PB16	*Lactobacillus paracasei* L9	*Lactobacillus paracasei* strain L9
	*Bifidobacterium longum* subsp. *longum*	*Bifidobacterium longum subsp. longum* JCM 1217
	*Lactobacillus acidophilus* strain *FS14*	*Lactobacillus acidophilus* strain FSI4
	*Bifidobacterium animalis* subsp. *lactis*	*Bifidobacterium animalis* subsp. *lactis* strain A6
PB17	*Lactobacillus rhamnosus* LOCK908	*Lactobacillus rhamnosus* LOCK908
	*Bifidobacterium longum* subsp. *longum*	*Bifidobacterium longum* subsp. *longum* JCM 1217
	*Lactobacillus plantarum* subsp. *plantarum*	*Lactobacillus plantarum* subsp. *plantarum* ST-III
	*Lactobacillus helveticus*	*Lactobacillus helveticus* DPC 4571
	*Lactococcus lactis* subsp. *lactis*	*Lactococcus lactis* subsp. *lactis* CV56
	*Lactobacillus acidophilus* FSI4	*Lactobacillus acidophilus strain* FSI4
	*Lactobacillus casei* subsp. *casei*	*Lactobacillus casei* subsp. *casei* ATCC 393
	*Lactobacillus delbrueckii* subsp. *bulgaricus*	*Lactobacillus delbrueckii* subsp. *bulgaricus* ATCC BAA-365
	*Bifidobacterium breve*	*Bifidobacterium breve* JCM 7017
	*Lactobacillus reuteri*	*Lactobacillus reuteri* JCM 1112
	*Streptococcus thermophilus*	*Streptococcus thermophilus* LMD-9
	*Bifidobacterium bifidum* BGN4	*Bifidobacterium bifidum* BGN4
	*Bifidobacterium animalis* subsp. *lactis*	*Bifidobacterium animalis* strain A6
	*Lactobacillus brevis*	*Lactobacillus brevis* KB290
	*Lactobacillus salivarius*	*Lactobacillus salivarius strain* JCM1046
PB18	*Bifidobacterium bifidum*	*Bifidobacterium bifidum* S17
	*Bifidobacterium longum*	*Bifidobacterium longum strain:* 105-A
	*Lactobacillus gasseri*	*Lactobacillus gasseri* ATCC 33323
PB19	*Lactobacillus rhamnosus* LOCK908	*Lactobacillus rhamnosus* LOCK908
	*Lactobacillus acidophilus* strain FSI4	*Lactobacillus acidophilus* strain FSI4
	*Bifidobacterium animalis* subsp. *lactis*	*Bifidobacterium animalis* strain A6

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
