# Peer review of "Viability and Composition Validation of Commercial Probiotic Products by Selective Culturing Combined with Next-Generation Sequencing"

_microorganisms, 2019, doi:10.3390/microorganisms7070188_

Round 1

Reviewer 1 Report

Dear Authors,

The subject raised in this manuscript is very important but before review this manuscript few basic methodological issues must be solved:

1. in the manuscript this is lack of information concerning products storage conditions, procedure of the products collection and methodology used by manufacturer in the quality control. Without such information, we don’t know what was in fact validated. Probiotics labelling process is very specific and official labelling requirements, very often limit the informative role of the label. In this study only information on labels were validated – this is interesting for food or sanitary authorities, but not readers of scientific journal,

2. I found few products without information about strains on the label. Such products don’t fulfil requirements for probiotics, which activity is strain specific, and therefore should not be included in this manuscript,

3. the readability of figures is unacceptable, the analysis of figures is impossible.

After including these important issues in the manuscript I would be happy to read it again.

Author Response

Dear Reviewer,

         Thank you for the opportunity to revise our manuscripts. The points raised by you has surely enhanced the quality of our manuscript. We have followed your suggestions and made the changes as suggested. We once again are thankful to you.

Response to Reviewer 1 Comments

Point 1: In the manuscript this is lack of information concerning products storage conditions, procedure of the products collection and methodology used by manufacturer in the quality control. Without such information, we don’t know what was in fact validated. Probiotics labelling process is very specific and official labelling requirements, very often limit the informative role of the label. In this study only information on labels were validated – this is interesting for food or sanitary authorities, but not readers of scientific journal,

Response 1: Thanks for your comments. To address the concerns raised in point 1, we have included an additional sub heading in materials and methods section of the manuscript. This sub heading (lines 79-84) provide the product information, storage and collection. However, it is not feasible to include the quality control measurements information used during the manufacturing of the products as it wasn’t provided in the products label.

Point 2: I found few products without information about strains on the label. Such products don’t fulfil requirements for probiotics, which activity is strain specific, and therefore should not be included in this manuscript.

Response 2: We appreciate your comment. Although the point raised here is valid, however, during the random collection of probiotic products samples, we came across few products that provided species-level information only. We thus added these few products in our analysis to validate our claim of the effectiveness of culture-based sequencing approach for strain level identification. This provides a proof about the effectiveness of our approach which we suggest could be used as a diagnostic tool for commercial probiotics qualitative assessment.

Point 3: The readability of figures is unacceptable, the analysis of figures is impossible.

Response 3: Thanks for the observation. The quality of the figures has been enhanced to make them comprehensible. We have also included figures both in the main manuscript and as a single separate file. The editorial office may make the files available to the reviewer upon request.

Reviewer 2 Report

Viability and validation of commercial probiotic products by selective culturing combined with next generation sequencing. By Mati Ullah. This study tried to highlight the qualitative and quantitative assessment approach for accurate measurement of commercial probiotic supplements. Compositional analysis of bacterial communities in supplements (probiotics) by metagenomics approach is recently published. However, combining the selective culturing provides a bit novelty. From quantitative assessment point of view this work is interesting and inclusion of selective culture combined with next-generation sequencing provides a good tool for further quality control especially in food industry. However, there are few concerns as following:

1.      The difference in labeled and determined CFU is in same log values (Table 2). Is this difference true or will affect the beneficial claim?

2.       Authors talk about low-quality reads of PB1 and PB3, but they have not mentioned the CFU in table 2. Did authors try to culture the bacteria of PB1 and PB3, if yes what was the CFU and difference in between labeled claim and detected CFU?

3.      Observed difference in compositional analysis compared to label in terms of different bacterial species is alarming. It does not matter whether the bacterial species is pathogenic or not therefore, Line 196-198, stating that ‘these discrepancies although can’t relate to any pathogenic or harmful effect of bacterial contents but are alarming for species and strain-specific targeted use of probiotic therapy’ can be avoided or reframed.

4.      It is not clear in materials and method whether author used only one batch and lot or different batches and lots of these commercial probiotics?

5.      It might be beyond the scope of this study but it would be great if authors can include some probiotic supplements that are being used in European, American or any other Asian country. This would enhance impact of current study.

Author Response

Dear reviewer, 

        Thank you for the opportunity to revise our manuscript. The suggestions you provided has surely enhanced the quality and clarity of our manuscript especially, the materials and methods section. We have thoroughly followed your suggestions/concerns and made the changes in revised manuscripts. We once again are thankful to you.

Response to Reviewer 2 Comments

Point 1: The difference in labelled and determined CFU is in same log values (Table 2). Is this difference true or will affect the beneficial claim?

Response 1: Thank you for your comment. The difference in labelled and determined CFU value is same log values. The determined CFU value is based on the average value of the three replicate of selected culturing plate count. More specifically, each bacterial content in a product was plated on culture specific growth medium plate in triplicate and the value obtained was the average of the three plate. This method is the standard CFU counting method and is widely used. We have also added a detail description about the calculation of CFU counts from culture plates in material and methods section (line 97-105). The reduction in CFU count is considered to effect the specific health benefits associated with a particular probiotic strain and do not fulfil the requirement of Probiotic defined by the International Scientific Association of Probiotics and Prebiotic.

Point 2: Authors talk about low-quality reads of PB1 and PB3, but they have not mentioned the CFU in table 2. Did authors try to culture the bacteria of PB1 and PB3, if yes what was the CFU and difference in between labelled claim and detected CFU?

Response 2: We appreciate your comments. However, the two products PB1 and PB3 were excluded from analysis because of the low quality sequencing reads. We have only mentioned these two products in result section (line 192-193) in order to clear the confusion about the numbering (PB2 –PB19) of the tested probiotic samples. These two products are not part of the current analysis.

Point 3: Observed difference in compositional analysis compared to label in terms of different bacterial species is alarming. It does not matter whether the bacterial species is pathogenic or not therefore, Line 196-198, stating that ‘these discrepancies although can’t relate to any pathogenic or harmful effect of bacterial contents but are alarming for species and strain-specific targeted use of probiotic therapy’ can be avoided or reframed.

Response 3: We appreciate these comments and as suggested by the reviewer, the line 196-198 has been removed.

Point 4: It is not clear in materials and method whether author used only one batch and lot or different batches and lots of these commercial probiotics?

Response 4: We have randomly collected and tested 17 different commercial probiotic products. Our analysis is based on one lot for each of the tested product. An additional heading in material and methods section (line 79-84) is included that provide the detail information of the tested products.  

Point 5:   It might be beyond the scope of this study but it would be great if authors can include some probiotic supplements that are being used in European, American or any other Asian country. This would enhance impact of current study.

Response 5: Thanks for your suggestion. The tested products were from both local and multinational manufacturers. However, to keep the anonymity of these products, each of the tested product was renamed prior to analysis.

Reviewer 3 Report

The current manuscript investigates on the claims made by probiotic manufacturers regarding the content of live microorganisms of their product. This investigation is very interesting and could serve as a diagnostic tool for the assessment of commercial probiotic products especially looking on the viability and composition of the microorganisms. One comment for the author is to improve the figures. It is not clear, and some can even be converted as tables.

Author Response

Dear reviewer, 

         Thank you for the opportunity to revise our manuscripts. The suggestion you provided about the quality of figures has surely enhanced the quality of our manuscript. We have added more clear images as per your suggestion. We once again are thankful to you.

Response to Reviewer 3 Comments

Point 1: The current manuscript investigates on the claims made by probiotic manufacturers regarding the content of live microorganisms of their product. This investigation is very interesting and could serve as a diagnostic tool for the assessment of commercial probiotic products especially looking on the viability and composition of the microorganisms. One comment for the author is to improve the figures. It is not clear, and some can even be converted as tables.

Response 1: Thanks for the observation. The quality of the figures has been enhanced to make them comprehensible. We have also included figures both in the main manuscript and as a single separate file. The editorial office may make the files available to the reviewer upon request.

Reviewer 4 Report

This study on the accurate labelling of probiotic products, both in terms of live cells and of bacterial strains, is timely and welcome. However, the article needs some major revisions.

MAJOR COMMENTS

English must be revised. For instance, lines 36-37 reads Additionally, it has been found effective…”, but the statement refers to probiotics, which is plural, and thus the line should read Additionally, they have been found effective…”.

CFU counting is known to produce significant variability, and authors only state that standard CFU counting method was used, without giving details.

There is a clear need to indicate: i) whether samples were surface plated or were plated by inclusion; ii) method used to obtain anaerobic conditions (e.g. did they use anaerobic jars with the Anaerocult ki?); iii) the number of replicates used, and iv) provide the CFU range for each product on Table 2, not only the average (they can either provide minimum, median and maximum values, or average and standard deviation). Finally, there is no indication of whether results from Table 2 correspond to the sum of the different selective media (which can both overestimate or underestimate counts depending on the true selectivity of the media, as well as their recovery capacity) or to the standard MRS plating, which can possibly underestimate counts in Bifidobacteria-containing products (those products could benefit from supplementing MRS with Cystein-HCl).

Throughout the article, the authors assume that bacterial contents not found during culture are non-viable, but strictly speaking, these bacteria should be labelled as non-culturable. The non-culturable status can arise from absence, cell death or from live cells switching to a non-culturable phenotype. A positive PCR result rules out absence, but does not allow to conclude whether the cells are dead or in a non-culturable state. Thus, non-viable should be changed to non-culturable, and this inability to differentiate between non-viable and non-culturable should acknowledged in the Discussion and the Abstract.

The paper would also benefit from the authors further elaborating on the procedure to precisely identify the strains from the sequencing data. i.e. how was NGS data processed to identify each strain. Did they compare against previously deposited genomes? If so, where did they download all the genomes data? Also, how did they design the strain-specific primers from NGS data? Which software and settings did they use?

The article uses two instances of PCR analysis: strain-specific PCR (described in lines 134-142, using the primers from Supplementary Table 1) and species-specific PCR for uncultured bacteria (described in lines 143-150). The paper would greatly benefit from clearly indicating in each instance in the text whether the authors refer to the strain-specific or the species-specific PCR. Also, for species-specific PCR (described in lines 143-150), the authors should clearly indicate if they used the original PCR conditions indicated in each cited paper (references 20 to 25) or a fix set of conditions for all PCRs.

Resolution of Figures 1, 2 and 3 must be greatly improved.

In Figures 2 and 3, were the reads assigned to each bacterial strain assembled into genomes? Otherwise, if using full genome sequencing, relative abundance of bacteria with larger genomes will be overestimated.

MINOR COMMENTS

Line 32: Change International Association of Probiotics and Prebiotic for International Scientific Association of Probiotics and Prebiotic.

Line 44: To keep coherence with the importance of strains stated by the authors in line 33, Id suggest this line to be rephrased to Strains of several bacterial species are used in…”

The paper would gain in clarity if a workflow or drawing of the culture-based sequencing approach was included.

Author Response

Dear reviewer,

         Thank you for your thorough review of our manuscript. The concerns you raised and the suggestions you provided will significantly enhance the quality and clarity of our manuscript. We took your concerns/suggestion seriously and made significant changes in our revised manuscript based on it. We tried our best to explain each point you raised in detail. We once again are thankful to you.

Response to Reviewer 4 Comments

Major Comments

Point 1: English must be revised. For instance, lines 36-37 reads “Additionally, it has been found effective…”, but the statement refers to probiotics, which is plural, and thus the line should read “Additionally, they have been found effective…”.

Response 1: Thank you for your comment. We have changed the mentioned lines “it has been found effectivetothey have been found effective” as suggested by the reviewer. Please see line 37. Additionally, we have carefully read the manuscript to ensure there are no major language issues.

Point 2: CFU counting is known to produce significant variability, and authors only state that “standard CFU counting method” was used, without giving details.

CFU counting is known to produce significant variability, and authors only state that “standard CFU counting method” was used, without giving details.

There is a clear need to indicate: i) whether samples were surface plated or were plated by inclusion; ii) method used to obtain anaerobic conditions (e.g. did they use anaerobic jars with the Anaerocult kit); iii) the number of replicates used, and iv) provide the CFU range for each product on Table 2, not only the average (they can either provide minimum, median and maximum values, or average and standard deviation). Finally, there is no indication of whether results from Table 2 correspond to the sum of the different selective media (which can both overestimate or underestimate counts depending on the true selectivity of the media, as well as their recovery capacity) or to the standard MRS plating, which can possibly underestimate counts in Bifidobacteria-containing products (those products could benefit from supplementing MRS with Cystein-HCl). 

Response 2: We appreciate your comments. A pour plate technique was used for culturing the bacterial contents. The culturing was carried in anaerobic incubator. The CFU counting was based on the average value of the bacterial colonies on three replicates of selective media plate to ensure maximum recovery for every bacterial species tested. The determined value represented in Table 2 is the average values obtained from three replicate of selective media for each bacterial species and is represented CFU/g of the product. For Bifidobacteria-containing products we have used MRS-NPNL which is one of the ideal medium and is widely reported to culture Bifidobacterial species. The information about the Culturing media is summarized in Table 1. Based on your suggestions, relevant sentences have been added to the Materials and methods section. Please see lines 97-105.

Point 3 : Throughout the article, the authors assume that bacterial contents not found during culture are non-viable, but strictly speaking, these bacteria should be labelled as non-culturable. The non-culturable status can arise from absence, cell death or from live cells switching to a non-culturable phenotype. A positive PCR result rules out absence, but does not allow to conclude whether the cells are dead or in a non-culturable state. Thus, “non-viable” should be changed to “non-culturable”, and this inability to differentiate between non-viable and non-culturable should acknowledged in the Discussion and the Abstract.

Response 3: Thanks for your comment. The term, “non-viable” has been changed to “non-culturable”, and the suggested acknowledgement has been made both in the Abstract (line 16) and Discussion sections (lines 279-286).

Point 4 : The paper would also benefit from the authors further elaborating on the procedure to precisely identify the strains from the sequencing data. i.e. how was NGS data processed to identify each strain. Did they compare against previously deposited genomes? If so, where did they download all the genomes data? Also, how did they design the strain-specific primers from NGS data? Which software and settings did they use?

Response 4: We appreciate these comments. Our Bioinformatics analysis of the sequencing data basically include the assembly, mapping and calculation of relative abundances of each bacterial species/strains in a corresponding probiotic product. The tool for assembly is mentioned in the text. Additionally, we developed an in-house database by downloading a comprehensive collections of normal gut bacteria from NCBI and processed our assembled data against this database to identify the bacterial contents of each probiotic product. The relative abundance of bacterial contents was determined by mapping the sequencing reads to these assembled bacterial genomes and represented as percentage (as several bacterial contents were present in single products for majority of the products). In summary, the following points provide answers to the queries raised.

·         The sequencing reads of NGS data were assembled and blast against the in-house developed database in our Linux server to identify the bacterial species/strain.

·         Yes, we have downloaded a comprehensive collections of normal gut bacteria from NCBI and identified each strain against these bacterial species/strain.

·         The data was downloaded to our Linux server to make an in-house database.

·         The primers were designed by Primer3 with the default settings (Please see line 152)

Point 5: The article uses two instances of PCR analysis: strain-specific PCR (described in lines 134-142, using the primers from Supplementary Table 1) and species-specific PCR for uncultured bacteria (described in lines 143-150). The paper would greatly benefit from clearly indicating in each instance in the text whether the authors refer to the strain-specific or the species-specific PCR. Also, for species-specific PCR (described in lines 143-150), the authors should clearly indicate if they used the original PCR conditions indicated in each cited paper (references 20 to 25) or a fix set of conditions for all PCRs.

Response 5:  Thanks for your comment. As suggested, we have revised the text and also indicated the species-specific and strain-specific PCR accordingly. For clarity, the validation of uncultured bacterial species was done with species-specific PCR reaction setting established previously without any modifications (Please see line 162) while strain-specific PCR was mainly used to validate our sequencing results (line 248).

Point 6: Resolution of Figures 1, 2 and 3 must be greatly improved.

Response 6: Thanks for your suggestion. The resolution of the Figures has been increased in compliance with the journal’s specifications. We included the figures in the main manuscript and as a separate file in three different formats (PDF, TIFF, and PNG) during submission. The editorial office can make the files available to the reviewer upon request.

Point 7: In Figures 2 and 3, were the reads assigned to each bacterial strain assembled into genomes? Otherwise, if using full genome sequencing, relative abundance of bacteria with larger genomes will be overestimated.

Response 7: The relative abundance of bacterial species was determined based on mapping of sequencing reads to the assembled genome. Hence it is not overestimated. We provided clarity on this in response 4.

Minor Comments

Point 1: Line 32: Change “International Association of Probiotics and Prebiotic” for “International Scientific Association of Probiotics and Prebiotic”.

Response 1: The change has been made as suggested (Please see line 32).

Point 2: Line 44: To keep coherence with the importance of strains stated by the authors in line 33, I’d suggest this line to be rephrased to “Strains of several bacterial species are used in…”

Response 2: The change has been made as suggested (Please see line 44).

Point 3: The paper would gain in clarity if a workflow or drawing of the culture-based sequencing approach was included.

Response 3: We appreciate your suggestion, however, in our opinion the culture-based sequencing approach is well explained in several steps in text and we feel that it is simple enough to understand by the reader without any workflow or drawing.

Round 2

Reviewer 1 Report

Dear Authors,

The manuscript was significantly improved. Because you have measured bacterial culturability, the title of your manuscript should be changed. I would propose as follows: “Results of selective culturing of commercial probiotic products combined with next-generation sequencing”. This title is fair and correct from scientific point of view. I think that you should conclude that manufacturers should inform customers (on label or website) about methodology used in the quality control, especially concerning bacterial viability and microbial purity.

Author Response

 Dear Reviewer, 

                          Thank you for review our manuscript. Your comments and suggestions are really helpful to improve the quality of our manuscripts. We once again are thankful to you.

Response to Reviewer 1 Comments

Point 1: The manuscript was significantly improved. Because you have measured bacterial culturability, the title of your manuscript should be changed. I would propose as follows: “Results of selective culturing of commercial probiotic products combined with next-generation sequencing”. This title is fair and correct from scientific point of view. I think that you should conclude that manufacturers should inform customers (on label or website) about methodology used in the quality control, especially concerning bacterial viability and microbial purity.

Response 1: Thank you so much for your comment. We appreciate the suggestion. As we have measured and analyzed the bacterial culturability and composition of commercial probiotic products. Hence we believe that the title of the current manuscript is simple and precise to understand the aim of the study.

 For the suggestion related to the methodology and quality control measurements, we have followed your suggestion and made the desired changes in the discussion (last paragraph) part of the manuscript (line 338-342)

Reviewer 4 Report

MAJOR COMMENTS

#1 In my previous review I asked the authors to provide information on the CFU range of each product. Table 2 must contain this information, be it in the form of Median plus Min and Max value, or in the form of Average plus standard deviation. Currently only the average value is reported. This is one of the key results of the article an in my opinion this article CANNOT be published without this information. This should be very simple to do, as the authors indicated they performed triplicates, and therefore, obtaining the minimum and maximum values or the standard deviation is straightforward. Also, the authors MUST indicate whether the triplicates were performed on the same dilution bank or on independent dilution banks for each replica, as in my experience this can strongly influence the variability of the results.

#2 In my previous review I asked the authors to provide more information on the NFG procedure (Point 4). They have provided some answers to me (below in Italics):

· The sequencing reads of NGS data were assembled and blast against the in-house developed database in our Linux server to identify the bacterial species/strain.

· Yes, we have downloaded a comprehensive collections of normal gut bacteria from NCBI and identified each strain against these bacterial species/strain.

· The data was downloaded to our Linux server to make an in-house database.

I believe these should be included in the manuscript text. Also, they should indicate the date of download of the NCBI databases to their servers the BLAST parameters

#3 I stand by my comment that the paper would gain in clarity if a workflow or drawing of the culture-based sequencing approach was included. I agree with the authors that the methods can be clear for microbiology specialists. However, the information of this article can be of high interest to physicians and nutritionists, which may easily get lost in the details of the multi-step methodology used in this article. Thus, adding a workflow will facilitate the understanding of the methodology to a wide range of potential readers which can really benefit from the work done.  

MINOR COMMENTS

Line #37: The sentence suggests probiotics have been proved to work in rheumatoid arthritis and urinary tract infections. This feels like an overstatement. The sentence should be rephrased to something like “some studies suggest that some probiotic strains could be effective in rheumatoid arthritis and urinary tract infections”

Line 155: Change “starin-specific” for “strain-specific”.

Line 158: Please, add reference for Primer3 software.

Author Response

Dear Reviewer, 

                        Thank you so much for your precious time and constructive comments and suggestions that will surely improve the quality and impact of our manuscript. In the second revision of the manuscript , we have made all the changes as suggested by you. We once again are thankful to you.

Response to Reviewer 4 Comments

MAJOR COMMENTS

Point 1:  In my previous review I asked the authors to provide information on the CFU range of each product. Table 2 must contain this information, be it in the form of Median plus Min and Max value, or in the form of Average plus standard deviation. Currently only the average value is reported. This is one of the key results of the article an in my opinion this article CANNOT be published without this information. This should be very simple to do, as the authors indicated they performed triplicates, and therefore, obtaining the minimum and maximum values or the standard deviation is straightforward. Also, the authors MUST indicate whether the triplicates were performed on the same dilution bank or on independent dilution banks for each replica, as in my experience this can strongly influence the variability of the results.

Response 1: Thank you for your comment. We really appreciate your suggestions. Based on your suggestions, you have given us two option to represent the determined CFU counts. We have followed your suggestion and measured the standard deviation. In the revised version of manuscript the determined CFU counts is in the form of average plus standard deviation. The triplicates were performed on same dilution bank. We have added this description in line 101-102.

Point 2:   In my previous review I asked the authors to provide more information on the NFG procedure (Point 4). They have provided some answers to me (below in Italics):

· The sequencing reads of NGS data were assembled and blast against the in-house developed database in our Linux server to identify the bacterial species/strain.

· Yes, we have downloaded a comprehensive collections of normal gut bacteria from NCBI and identified each strain against these bacterial species/strain.

· The data was downloaded to our Linux server to make an in-house database.

I believe these should be included in the manuscript text. Also, they should indicate the date of download of the NCBI databases to their servers the BLAST parameters.

Response 2: We have thoroughly followed your suggestions in our revised manuscript and added these description to the manuscripts text (line 142-150). Additionally, we have added description about the date of download of data from NCBI (line146-147) along with the blast parameters (line 147-148).

Point 3: I stand by my comment that the paper would gain in clarity if a workflow or drawing of the culture-based sequencing approach was included. I agree with the authors that the methods can be clear for microbiology specialists. However, the information of this article can be of high interest to physicians and nutritionists, which may easily get lost in the details of the multi-step methodology used in this article. Thus, adding a workflow will facilitate the understanding of the methodology to a wide range of potential readers which can really benefit from the work done.  

Response 2: We appreciate the point raised here. As suggested, we have included a workflow (figure 1 in the revised version of the manuscript) that show the various steps involved in the culture-based sequencing approach of probiotic products analysis.

MINOR COMMENTS

Line #37: The sentence suggests probiotics have been proved to work in rheumatoid arthritis and urinary tract infections. This feels like an overstatement. The sentence should be rephrased to something like “some studies suggest that some probiotic strains could be effective in rheumatoid arthritis and urinary tract infections”

Reply. The sentence was rephrased as suggested (line36)

Line 155: Change “starin-specific” for “strain-specific”.

Reply. The spelling was corrected (line153)

Line 158: Please, add reference for Primer3 software.

Reply. Reference was added for Primer3 software (line 156, reference 21).
